# Toward Gender Equality in Education—Teachers' Beliefs about Gender and Math

**Jana Lindner [1],* [ID], Elena Makarova [1] [ID], Deborah Bernhard [2] and Dorothee Brovelli [3]**

[1] Institute for Educational Sciences, University of Basel, 4132 Muttenz, Switzerland; elena.makarova@unibas.ch

[2] Institute of Professional Research and Competence Development, University of Teacher Education St. Gallen, 9000 Sankt Gallen, Switzerland; deborah.bernhard@phsg.ch

[3] Institute for Education in Science and Social Studies, University of Teacher Education Lucerne, 6003 Luzern, Switzerland; dorothee.brovelli@phlu.ch

* Correspondence: jana.lindner@unibas.ch

**Abstract:** Math has a strong gender-related image, even among teachers. As teachers hold beliefs about their work, their role, their subject, and their students, they shape girls' and boys' mathematical beliefs and attitudes. Research during the past 20 years has shown that teachers' gender beliefs about mathematics significantly favor boys, thereby reinforcing girls' low math ability self-concept. Still, there is a lack of studies that examine teachers' gender-related beliefs based on their underlying assumptions. Our study provides the first empirical evidence of the relationship between general gender stereotypes and math stereotypes. To this end, we used partial correlation and MANCOVA to analyze data from an online survey in 2019/2020 conducted in Switzerland (195 women, 80 men) as part of a cross-cultural comparison study. We therefore created a differentiated profile of prospective teachers by examining their beliefs about their self-image, their image of men and women in society, their essentialist and gender role ideology beliefs, and their math stereotypes. Then, we linked prospective teachers' beliefs about gender (based on 48 characteristics) to their beliefs about mathematics and about girls' and boys' competencies in math. The extensive analysis provides knowledge about prospective teachers and is particularly important for teacher education.

**Keywords:** beliefs; gender stereotypes; math stereotypes; prospective teachers; self-views; communality; agency; weakness; dominance

## 1. Introduction

### 1.1. Gender Disparities Related to Math

Fundamental mathematical knowledge, gained in the early years of (pre-)school, is essential throughout and in various areas of life, as it can shape logic and thinking development, problem-solving skills, and creativity [1,2]. Yet, by the end of elementary school, many female students have internalized their teachers' gender biases regarding talent ascription in math, as they perceive themselves as being less talented than boys [3].

With respect to the development of math-related competencies, large-scale studies such as PISA and TIMSS have reputedly highlighted gender differences in competencies related to STEM (science, technology, engineering, and mathematics) subjects. The Program for International Student Assessment (PISA), launched and conducted by the OECD, shows that in 21 of the 37 OECD countries, boys outperformed girls in math, but there were no significant gender differences in the 13 OECD countries, and in three OECD countries, Finland, Norway, and Iceland, girls showed higher mathematical competence than boys [4] (p. 199). The international school achievement survey, TIMSS (Trends in International Mathematics and Science Study) measured 4th- and 8th-grade students' basic mathematical understanding [5]. On average, 4th-grade boys scored higher than girls in nearly half of the participating countries—a total of 27—in 2019. In four countries, the differences were in favor of girls, and in the remaining 27 countries, the average mathematic achievement

was equally balanced [5] (p. 25). In the cohort of 8th graders, 26 of the 39 participating countries showed a balanced picture of girls' and boys' mathematics achievement. On average, girls performed better in math in seven countries and boys performed better in six countries [5] (p. 164).

It is important, though, to take a more differentiated look at female and male students' performance to avoid overgeneralization of gender differences in such tests: Among 4th graders, in the area of "number" (included pre-algebra), girls performed better in 3 countries and boys in 25 countries. In the area "measurement and geometry", girls achieved better results in 3 countries and boys in 26 countries. In the area of "data", boys performed better in 11 countries and girls in 8 countries [5] (p. 68). Focusing on the sub-disciplines tested among 8th graders, the following picture emerges: In the area of "number", gender-specific performance differences in favor of boys became evident in 14 countries whereas girls achieved better scores than boys in 4 countries. Girls' performance was particularly strong in "algebra". Here, the girls performed better than boys in 16 countries. Boys, on the other hand, did not score higher than girls in algebra in any of the participating countries. Girls also achieved better results in "geometry" in seven countries and boys in three countries. In the area of "data and probability", boys were ahead of girls in nine countries, and girls outperformed boys in seven countries [5] (p. 206). Consequently, girls did not generally perform significantly worse than boys. Girls in the 8th grade performed exceptionally well in algebra and geometry compared to boys. Moreover, gender differences in average performance were small. In 4th grade, gender equity in terms of math achievement was 46.6%, with nine countries having closed the gap favoring boys since 2015 [5] (p. 98). In 8th grade, 66.7% of girls and boys performed equally well in math on average [5] (p. 164). On closer examination of those large-scale studies, the belief that competencies in math must inevitably be linked to one gender is untenable and should be questioned (e.g., [6]).

Although gender differences in math performance vary by country and age, boys' and girls' career paths diverge before the age of 15, when career decisions are typically made. For example, 15-year-old boys are, on average, more than twice as likely as girls to work as engineers, scientists, or architects [7] (p. 107), but fewer than 0.5% of girls in OECD countries aspire to become ICT professionals compared to boys (5%) [8].

Additionally, gender segregation in career choices proves to be very stable on an international level. According to several reports that address gender equality, that is, the European Commission's "Report on Gender Equality in the EU" [9] and the Global Gender Gap Report of the World Economic Forum [10], young women remain underrepresented in STEM. The underrepresentation of women is particularly severe in mathematics-intensive science fields, such as geosciences, engineering, economics, mathematics, computer science, and physics [11].

Taking these facts into account, the relevance of addressing teachers' gender-related beliefs about math becomes clear, as they play an important role in shaping students' mathematical beliefs and attitudes [12–15]. A low math ability self-concept negatively affects students' future career choices in STEM [16–19]. Moreover, a strong gender stereotype regarding math and science among female students decreases the likelihood that young women will enroll in STEM fields at university [20].

### 1.2. Beliefs about Gender and Gender Roles

Teachers and prospective teachers hold beliefs about their work, their role, their subject, and their students [21] (p. 314). Beliefs can be defined as mental constructs consisting of subjective knowledge and emotions based on individual experiences [22]. Therefore, they can be differentiated from factual, empirically validated knowledge [21]. Nevertheless, beliefs are culturally shared, as they provide structure and ensure a sense of belonging to a group [21]. However, because beliefs work on an individual level and on a sociocultural level, inconsistent beliefs can coexist within belief systems [23].

The concept of beliefs is difficult to distinguish from other psychological concepts, such as attitudes. Pajares [21] (p. 209) points out that beliefs "[ . . . ] travel in disguise

and often under alias-attitudes, values, judgments, axioms, opinions, ideology, perceptions, conceptions, conceptual systems, preconceptions, dispositions, implicit [ . . . ], explicit [ . . . ], [and] personal theories, [ . . . ] to name but a few that can be found in the literature".

Gender-related beliefs about masculine and feminine characteristics, as well as notions of appropriate roles for women and men, are part of a belief system. The belief system prescribes that what is female cannot be male and vice versa [24]. Following this dichotomous logic of gender, men and women seem to differ diametrically in their characteristics, physical attributes, abilities, interests, needs, and the behaviors they display. The gender belief system creates the impression of stable gender roles and hence, unchanging gender differences as well as inherently coherent representations of gender: "People expect others to fit into a relatively stable set of gender roles, traits, and physical attributes, generally believing, for example, that a person who is either masculine or feminine in one aspect of behavior is similarly masculine or feminine in other aspects of behavior" [25] (p. 948).

Beliefs about gender are closely linked to gender stereotypes, as they have a stereotypical character. Gender stereotypes include culturally shared ideas, generalized beliefs, and expectations about gendered characteristics and behaviors [26]. In accordance with common gender stereotypes, gender-specific differences in personality traits can be distinguished: They have descriptive and prescriptive components, that is, characteristics and behaviors which men and women typically exhibit and those that are expected for and intended by men and women [27]. Prescriptive stereotypes can be positively afflicted and hence, desirable, for example, that women should be communal and men agentic. Opposite or proscriptive stereotypes that are not desirable for one gender but tolerable for the other include dominance for women and weakness for men [28].

Gender stereotypes have an underlying explanation or essence that connects their attributes, thus establishing differences between groups and supporting dispositional inferences [29]. The underlying essence can explain gender stereotypes as either biological or shaped by social factors [30].

On one hand, beliefs about gender and gender roles can be essentialist. In fact, gender categories are the most essential of social categories [31]. Essentialist-based beliefs present gender as natural, historical, and immutable [32]. Essentialist gender beliefs explain differences between men and women based on evolutionary biological hypotheses. According to such beliefs, genetically predisposed gender-typical preferences and behaviors have developed in the process of evolution, and they express themselves in psychological and social areas. Therefore, the early hunter-gatherer culture, with its strong gender-based division of labor, was responsible for the development of gender-specific attributes and abilities [33]. Hunting required strong spatial skills and, over evolutionary time, reinforced this ability in males [34].

On the other hand, psychological gender differences can also be explained with reference to culture-specific characteristics and cross-cultural commonalities. Accordingly, physically determined and hence, universal differences between men and women, lead to a gender-specific division of labor and a specific gender role ideology based on stereotypes [35]. Social role theory [36] states that gender roles, or gender-typical behavior, are learned from observing men and women in everyday contexts. Girls and boys anticipate explicit and implicit social expectations of learned gender roles, according to which, women should prioritize household maintenance and childrearing, and men should prioritize their careers [37]. This system leads to distinct social roles and role-bound activities, and hence, the reproduction of gender stereotypes in career choices [38,39].

In the present study, we aimed to show to what extent stereotypical beliefs affect prospective teachers and, in this context, which evolutionary- or socioculturally-based beliefs about gender and gender roles exist among them. In this regard, the fact that we conducted the study in Switzerland is important, for Switzerland is one of the 10 countries with the most advanced implementation of gender equality [40], however, it was also one of the last European countries to grant women full civil rights in 1971. In rural areas, some women even had to wait until 1990 for women's suffrage to achieve gender

equality [41]. Therefore, the urban and rural socialization experiences of the prospective teachers surveyed were also likely to be significant in shaping beliefs, which we will address in the analysis.

First, we will present a differentiated profile of prospective teachers' beliefs about gender. Second, we will illustrate the prospective teachers' gender-related beliefs about math. Finally, we will link the prospective teachers' stereotypical beliefs about gender to their gender-related beliefs regarding math.

### 1.3. The Gender-Related Image of Math

Research has shown that mathematics and science subjects have a gender-related image [42,43]. Youth associate mathematics, chemistry, and physics with the male gender [43]. Among young women and men, mathematics, compared to chemistry and physics, has the strongest association with masculinity [20]. However, even among teachers, math has a strong male connotation [43–45]. The "math male stereotype" has also been demonstrated in various studies using the Implicit Association Test (IAT) to reveal implicit stereotypes whereas math and science were more associated with the male gender than female gender [46–48]. IAT-based results also show that gender-related beliefs about math have a negative impact on women and their career opportunities in STEM, with male and female employers associating women less with mathematics than men [49].

### 1.4. Teachers' Gender-Related Beliefs about Math

Empirical evidence from studies conducted in various countries have consistently shown that teachers have gender-stereotyped beliefs regarding their students' competencies and achievement in math (for a review, see [50]). The prevailing gender bias in math was expressed in teachers' performance assessments, in which boys' performance is rated better, despite boys' and girls' comparable achievement: As early as kindergarten, teachers estimated girls' mathematical ability as being lower than that of boys with similar achievements and learning behaviors [51]. In addition, 1st-grade teachers in the US attributed being more logical, competitive, and independent to their best male students more often than to their best female students in math [12]. Among 3rd to 5th graders in Germany, boys were also considered to have more logical thinking ability and stronger mathematical skills whereas math was considered more difficult for girls [52,53]. Teachers who have internalized stronger gender stereotypes also exhibited stronger gender stereotypes regarding their students' math ability [52,53].

Gender-related beliefs about math have also been demonstrated among secondary school teachers, who generally rated math as more difficult for 10th-grade girls than equally achieving boys [54]. Boys' success in math tended to be attributed to talent and natural mathematical ability as well as greater interest in math [3,12,51,55–58]. The belief that math requires an innate ability was more prevalent among middle school math teachers than elementary school math teachers and was also more likely to be found among teachers who believe that math requires talent, which they believe girls lack [59].

Girls' success in math was perceived as a result of greater effort whereas failure was interpreted as a result of a lack of ability [12,52]. Jaremus et al. [60] showed that teachers discursively constructed the candidates of mathematics courses as male people in interviews. By means of naturalistic arguments claiming that students were more talented in math because they have the necessary intellect, girls were excluded and gender stereotypes were further reinforced. Although the research results presented so far refer to in-service teachers, a similar picture emerged in studies on student teachers, whose judgments were also biased by gender [30,61,62]. Prospective teachers with little teaching experience assumed that girls will perform worse in mathematics than boys [62]. Nürnberger et al. [30] showed that preservice teachers had a stronger gender-stereotyped belief that math is for boys and languages are for girls. Copur-Gencturk et al. [59] found that compared to less experienced teachers, more experienced teachers believed less strongly that mathematical ability was malleable. However, the study's authors showed that most of the 382 elementary

and middle school teachers surveyed had low levels of agreement with gender-based stereotypes about math, according to which mathematical ability is innate. Instead, the teachers shared the view that hard work and dedication were crucial for success in math [59]. At the same time, a few other studies have shown contradictory results, revealing no gender bias among teachers in terms of gender-related misconceptions and assessment of students' math abilities [63,64].

Summarizing the state of research, the vast majority of studies have shown significant gender-related beliefs about math among teachers in favor of boys. Still, there is a lack of studies on teachers' gender-related beliefs based on their underlying assumptions. Moreover, we have no empirical evidence of a link between general gender stereotypes and gender-related stereotypes about math. Therefore, we aim to enrich the scientific discourse by answering the following three questions:

(1) What general beliefs do prospective teachers have about gender?
(2) What beliefs do prospective teachers have about math and female and male students' competencies in math?
(3) How are prospective teachers' general beliefs about gender related to their beliefs about math?

Our study's results will serve as a groundwork for further research on how teachers' beliefs regarding gender differences relate to the development of gender differences in students' beliefs, learning, and achievement. Furthermore, our findings give an indication for research on the relationship between gender-based differences in teachers' beliefs, their instruction, and the decisions they make in the classroom.

## 2. Materials and Methods

### 2.1. Participants

We conducted the study between October 2019 and March 2020 via an online survey in Switzerland. The Swiss study was part of the cross-cultural comparative study "Towards Gender Harmony—Understanding the Relationship between Masculinity Threat and Gender Equality Across Cultures", which the University of Gdańsk initiated to examine psychological traits attributed to men and women and thereby, to analyze gender stereotypes cross-culturally and cross-nationally, addressing precarious manhood beliefs and ambivalent sexism [65].

The Swiss sample originally consisted of 195 prospective female teachers (69.89%), 80 prospective male teachers (28.67%), and 2 participants (0.72%) who indicated that they did not identify with any gender. Likewise, two participants (0.72%) stated that they preferred a self-definition instead of the given options, "man" and "woman". Due to the low number of responses in these categories, we focused our analysis on the binary gender categories "man/male" and "woman/female".

We integrated three questions designed as attention controls in the online survey. We excluded five people from the analysis because they answered these control questions incorrectly. Regarding age, we excluded six extreme outliers (aged 50 and older) in the analysis to avoid a potential bias. On average, the prospective teachers were 25 years old ($Min$ = 17.0, $Max$ = 42.0, $M$ = 24.7, $SD$ = 5.00), with women being younger ($M$ = 23.8, $SD$ = 4.46) than men ($M$ = 26.9, $SD$ = 5.57). As the ages ranged from 17 to 42, we can assume that the data reflected the beliefs of two generations. Therefore, we included age in the analysis as a demographic control variable.

In addition, we used the place where a participant grew up as a second control variable. One hundred and eight (43.0%) participants grew up in the country or a small town, and 143 (57.0%) participants grew up in a city. One hundred and ninety-one (76.1%) prospective teachers were seeking to obtain a teaching license in math, and 60 (23.9%) participants were pursuing a teaching license in a subject other than math.

*2.2. Procedure*

We recruited participants at four universities of teacher education in the German-speaking part of Switzerland. The institutions sent a link to the online survey to their students, who in turn gave informed consent to participate. Participation in the survey was anonymous and voluntary. We conducted a pretest in lectures at the University of Basel in September 2019 and revised the questionnaire accordingly for the main survey.

*2.3. Instruments*

To investigate general beliefs about gender and, in a further step, prospective teachers' beliefs about math and about female and male students' competencies in math, we used a multidimensional framework. We measured general beliefs about gender using two constructs, the "self-construal" and the "prescription/proscription women" scales as well as the "prescription/proscription men" scale. On one hand, we intended to determine how the participating women and men perceived themselves and which characteristics they ascribed to ("*Rate the extent to which each of the traits describes you personally*") and, on the other hand, how the prospective teachers perceived women and men, and which characteristics they believed were desirable in each gender in society ("*How desirable is it in your society for a woman/man to possess each of the following traits?*").

Both constructs contained 48 identical items, and we asked participants to indicate to which extent 12 agentic traits, 12 communal traits, 12 weakness traits, and 12 dominance traits described them on a seven-point Likert scale of 1 ("*does not describe me at all*") to 7 ("*describes me well*") and, simultaneously, to what extent the same traits are desirable in men and women in society on a scale of 1 ("*not at all desirable*") to 7 ("*very desirable*"). We selected the traits listed in Table 1 from studies on prescriptive gender stereotypes [28,66–68]. Principal component analyses confirmed the construct's four-factor structure. We employed four factors for the "self-construal", "pre-/proscription women", and "pre-/proscription men" scales: (i) communality, (ii) weakness, (iii) dominance, and (iv) agency.

**Table 1.** Items of the scales "self-construal", "pre-/proscription women", and "pre-/proscription men".

| Communality | Weakness | Dominance | Agency |
|---|---|---|---|
| Items | Items | Items | Items |
| compassionate (1) | worrying (1) | demanding (1) | decisive (1) |
| helpful to others (2) | weak (2) | controlling (2) | ambitious (2) |
| sympathetic (3) | timid (3) | bossy (3) | competitive (3) |
| understanding of others (4) | submissive (4) | dominant (4) | competent (4) |
| sensitive (5) | fearful (5) | intimidating (5) | confident (5) |
| soft-hearted (6) | cowardly (6) | feels superior (6) | has leadership abilities (6) |
| aware of others' feelings (7) | dependent (7) | forceful (7) | efficient (7) |
| cooperative (8) | infantile (8) | dictatorial (8) | determined (8) |
| devoted to others (9) | uncertain (9) | aggressive (9) | courageous (9) |
| trusting (10) | approval seeking (10) | stubborn (10) | active (10) |
| warm (11) | subordinates self to others (11) | arrogant (11) | capable (11) |
| supportive (12) | insecure (12) | boastful (12) | independent (12) |

We used the scale "teachers' gender stereotype toward mathematics" to measure prospective teachers' beliefs about math and about female and male students' competencies in math [69]. The participants indicated the degree to which they agreed or disagreed with each statement using a five-point Likert scale from 1 ("*disagree strongly*") to 5 ("*agree strongly*"). For the purpose of this study, we used the four following subscales.

(i)　Environment (four-item scale) examines teachers' beliefs about how peers and parents perceive students in mathematics (i.e., "*Compared to girls, boys are seen more competent in mathematics by their parents*").

(ii) Career (four-item scale) examines teachers' beliefs about students' career choices (i.e., "*Boys are more interested in careers that require mathematical abilities than girls are*").

(iii) Competence (six-item scale) examines teachers' beliefs about students' mathematical knowledge, ability, and attitudes (i.e., "*Boys understand mathematical concepts more easily than girls do*").

(iv) Attribution of success (three-item scale) examines teachers' beliefs about the reasons for students' achievements (i.e., "*Compared to girls, boys mostly increase their mathematical achievement because of the support of their teachers*").

A principal component analysis for the 17 items confirmed the four factors: (i) environment, (ii) gender appropriateness of careers, (iii) competence, and (iv) attribution of success. To analyze essentialist beliefs regarding gender, we applied the scale "gender essentialism" [32]. We asked participants to rate three statements (i.e., "*Men and women have different abilities*") on a seven-point Likert scale from 1 ("*strongly disagree*") to 7 ("*strongly agree*"). We calculated Cronbach's alpha coefficient to measure the internal consistency of all applied instruments, showing that the scales formed are reliable, as indicated in Table 2 below ($\alpha > 0.65$ [70]).

**Table 2.** Reliability of the applied scales.

| Scales | Subscales | Cronbach's Alpha | Number of Items | N | Mean | SD |
|---|---|---|---|---|---|---|
| Self-construal | Communality | 0.86 | 12 | 255 | 5.57 | 0.66 |
| | Weakness [1] | 0.85 | 12 | 255 | 3.28 | 0.95 |
| | Dominance | 0.87 | 12 | 255 | 2.95 | 0.91 |
| | Agency [2] | 0.83 | 12 | 255 | 4.91 | 0.79 |
| Pre-/Proscription Women | Communality | 0.88 | 12 | 254 | 6.06 | 0.64 |
| | Weakness [3] | 0.89 | 12 | 254 | 2.67 | 1.07 |
| | Dominance | 0.87 | 12 | 254 | 2.35 | 0.77 |
| | Agency | 0.91 | 12 | 254 | 5.41 | 0.92 |
| Pre-/Proscription Men | Communality | 0.93 | 12 | 249 | 5.47 | 1.01 |
| | Weakness [4] | 0.79 | 12 | 249 | 1.92 | 0.64 |
| | Dominance [5] | 0.89 | 12 | 248 | 2.56 | 1.01 |
| | Agency | 0.88 | 12 | 249 | 5.92 | 0.67 |
| Teachers' Gender Stereotypes toward Mathematics | Environment | 0.81 | 4 | 244 | 2.75 | 0.89 |
| | Career [6] | 0.75 | 3 | 233 | 3.22 | 0.92 |
| | Competence [7] | 0.93 | 5 | 247 | 1.85 | 0.89 |
| | Attribution of success | 0.71 | 3 | 232 | 2.01 | 0.81 |
| Gender Essentialism | | 0.68 | 3 | 254 | 3.91 | 1.38 |

Note: Eight items were excluded: [1] self-construal_weakness_8 (infantile) with F2 = −0.06; [2] self-construal_agency_9 (courageous) with F4 = 0.12; [3] prescription_women_weakness_8 (infantile) with F2 = −0.16; [4] prescription_men_weakness_1 (worrying) with F4 = 0.20; [4] prescription_men_weakness_8 (infantile): rotation was not possible; [5] prescription_men_dominance_1 (demanding) with F3 = 0.13; [6] career_7 (Boys are more suited than girls are to work in engineering branches) with F2 = −0.24; [7] competence_11 (Boys are more likely than girls are to believe they can be successful in mathematics) with F1 = 0.12.

The present study aimed to show the extent to which prospective teachers' gender-related beliefs are based on underlying, evolutionary-, or socioculturally-based assumptions. Hence, in addition to the "gender essentialism" scale, we used an item to analyze the gender (role) ideology that reflects what roles are prioritized in terms of sharing paid work, housework, and childcare ("*Looking at your own future, what will you prioritize?*"). Participants could choose between 0 ("*having family*") and 10 ("*having career*") by using a slider. For a better comparison, we divided the participants' answers into three groups: Those who prioritize family, those who prioritize career, and those who equally prioritize family and career.

*2.4. Statistical Analyses*

To answer the first research question regarding the general beliefs which prospective teachers have about gender, we performed multivariate analyses of covariance (MAN-

COVA) by including gender and "gender ideology" as independent variables, as well as demographic controls including age and the area of growing up. To examine the extent to which there is a significant relationship between female and male prospective teachers' essentialist-based beliefs, we conducted a *t*-test for independent samples using "gender essentialism" as a test variable.

Then, we performed partial correlation by using scales that analyze which characteristics prospective teachers believe to be desirable or undesirable for men and women in society, controlling for demographic variables (e.g., gender, age, area of growing up). To obtain a clearer profile of prospective teachers' beliefs about gender, we formed graphical indices using line graphs and compared the sample.

To answer the second research question regarding what beliefs prospective teachers have about math and about female and male students' competencies in math, we performed MANCOVA, using the four dimensions of the scale "teachers' gender stereotypes toward mathematics" as dependent variables, including the factors of gender and subject as independent variables, and controlling for demographic variables (e.g., age, area of growing up).

Finally, we answered the third research question of how prospective teachers' general beliefs about gender are related to prospective teachers' beliefs about math by performing partial correlation, using variables of the scales "pre-/proscription women", "pre-/proscription men", and "teachers' gender stereotypes toward mathematics", controlling for demographic variables (e.g., age, area of growing up).

We tested the assumptions for MANCOVA by using Shapiro–Wilk test, Levene's test (centers at the sample mean), Box's M test, and Mahalanobis distances. We checked the assumptions for the *t*-test and for the partial correlation. We analyzed the results presented below with SPSS 27.0.1.

## 3. Results

### 3.1. General Beliefs about Prospective Teachers' Gender

Concerning prospective teachers' general beliefs about gender, the MANCOVA found statistically significant differences between gender on communality (scale "self-construal"), as shown in Table 3. There was homogeneity of covariances, as assessed by Box's M test ($p > 0.001$), as well as homogeneity of the error variances ($p > 0.05$), as assessed by Levene's test. The analysis of the mean values indicated that women have stronger beliefs about their own gender regarding communality: Prospective female teachers agree more strongly than men do that those communal traits describe them well ($p < 0.001$). Other traits such as weakness, dominance, and agency showed no statistically significant differences.

With respect to prospective teachers' beliefs about gender roles or gender ideology, MANCOVA indicated a statistically significant difference between gender ideology on "communality" and "dominance" (Table 3). In terms of gender ideology beliefs ($n = 141$), most prospective teachers prioritize family and career equally. The second largest group represents those who prioritize family ($n = 76$). The smallest group prioritizes having a career ($n = 32$).

The post hoc test results showed that prospective teachers who equally prioritize family and career ascribe themselves as significantly more communal and have fewer dominant traits than those who have career-oriented gender ideology beliefs (Table 3). Prospective teachers who aspire to have a family also ascribe significantly less dominant traits to themselves than those prospective teachers who prioritize having a career. When controlling for the area of growing up (used as a factor) and age (used as a covariate), the MANCOVA showed no significant results.

**Table 3.** Self-descriptions related to gender, gender ideology, and controlling for growing up area and age (from left to right: means, MANCOVA, pairwise comparison for factor gender ideology, partial eta squared).

| | Gender (G) [a] | | Gender Ideology (GI) [a] | | | Growing Up Area (GA) [a] | | G | GI | GA | Age [b] | GI Pairwise Comparison [c], p | Partial η² [d] |
|---|---|---|---|---|---|---|---|---|---|---|---|---|---|
| | 1 | 2 | 1 | 2 | 3 | 1 | 2 | F, p df | F, p df | F, p df | F, p df | | |
| Communality | 5.67 | 5.29 | 5.54 | 5.63 | 5.31 | 5.60 | 5.53 | 14.11 *** | 3.48 * | 1.39 ns | 0.48 ns | 2 > 3 * | 10% |
| | 179 | 70 | 76 | 141 | 32 | 108 | 141 | 1/236 | 2/236 | 1/236 | 1/236 | | |
| Weakness | 3.35 | 3.11 | 3.34 | 3.22 | 3.42 | 3.32 | 3.26 | 1.94 ns | 0.33 ns | 1.16 ns | 1.92 ns | ns | 4% |
| | 179 | 70 | 76 | 141 | 32 | 108 | 141 | 1/236 | 2/236 | 1/236 | 1/236 | | |
| Dominance | 2.88 | 3.13 | 2.79 | 2.92 | 3.44 | 3.00 | 2.91 | 1.37 ns | 5.04 ** | 0.30 ns | 2.41 ns | 3 > 2 * | 8% |
| | 179 | 70 | 76 | 141 | 32 | 108 | 141 | 1/236 | 2/236 | 1/236 | 1/236 | 3 > 1 ** | |
| Agency | 4.89 | 4.92 | 4.88 | 4.94 | 4.77 | 4.88 | 4.92 | 0.88 ns | 0.94 ns | 0.43 ns | 2.60 ns | ns | 5% |
| | 179 | 70 | 76 | 141 | 32 | 108 | 141 | 1/236 | 2/236 | 1/236 | 1/236 | | |
| Results MANCOVA, Pillai's Trace | | | | | | | | 3.87 ** 4/944 | 2.32 * 8/944 | 0.97 ns 4/944 | 1.38 ns 4/944 | | |

Scale: 1 = does not describe me at all–7 = describes me well; [a] mean values and number of cases of the factor level (G): 1 = Women; 2 = Men; (GI): 1 = Family; 2 = Equal Priorities; 3 = Career; (GA): 1 = Urban; 2 = Rural; [b] Covariate; [c] Bonferroni adjusted; [d] Effect size showing percentage of variance associated with an effect; * $p < 0.05$; ** $p < 0.01$; *** $p < 0.001$; ns = not significant.

We performed a *t*-test for independent samples, using "gender essentialism" as a test variable to examine the extent to which there was a significant relationship between essentialist-based beliefs of female and male prospective teachers. According to the Shapiro–Wilk test, there was no normal distribution ($p$ = <0.05), but because the *t*-test is a robust test against the normality assumption [71], we continued the analysis using the *t*-test for unequal variances (Welch test). There were no outliers, and the homogeneity of variance assumption was met according to Levene's test ($p > 0.05$). There was no statistically significant difference between gender essentialist-based beliefs and male and female prospective teachers, t(125.53) = $-0.33$, $p = 0.746$. The mean values for men and women were almost identical: Female prospective teachers' essentialist-based beliefs were just slightly stronger than those of male prospective teachers (women $n = 181$, $M = 3.93$, $SD = 1.36$; men $n = 73$, $M = 3.88$, $SD = 1.45$).

*3.2. General Beliefs about Gender in Society*

After examining how male and female prospective teachers differed in terms of their beliefs about their own gender, as well as in terms of their gender ideological and gender essentialist beliefs, the study also analyzed the beliefs about pre- and proscriptive gender stereotypes which prospective teachers hold by performing a partial correlation. According to the Shapiro–Wilk test, there was no normal distribution ($p$ = < 0.05). Therefore, we used the nonparametric Spearman's rank-order correlation, controlling for demographic variables (e.g., gender, age, and area of growing up).

When correlating variables of the scales "pre-/proscription women" and "pre-/proscription men" among and with each other, the results indicated predominantly weak to moderate, statistically significant correlations (Table 4): Positive correlations showed that the degree of un-/desirability of one trait related to the degree of un-/desirability of the correlated trait, for example, with communal traits in men and dominant traits in women ($r_s$ = 0.131). Negative correlations indicated that the desirability degree of traits is associated with a decreasing degree of desirability of negatively correlated traits, likewise vice versa. There were, for example, negative correlations between communal traits in women and weak traits in men ($r_s$ = $-0.329$) or dominant traits in men and agentic traits in women ($r_s$ = $-0.385$). Yet, stronger but moderate correlations ($r_s \leq 0.5$) became evident in four cases, namely between weak traits in men and dominant traits in women ($r_s$ = 0.554), dominant traits in men and weak traits in women ($r_s$ = 0.554), agentic traits in men and communal traits in women ($r_s$ = 0.543), and communal traits in men and agentic traits in women ($r_s$ = 0.707).

**Table 4.** Partial correlation matrix (using Spearman's rank-order correlations): The lower-left correlations are unadjusted, and the upper-right correlations are partial correlations adjusted for gender, age, and area of growing up.

| | | Women Communality | Women Weakness | Women Dominance | Women Agency | Men Communality | Men Weakness | Men Dominance | Men Agency |
|---|---|---|---|---|---|---|---|---|---|
| Women Communality | $r_s$ | 1.000 | 0.096 | −0.238 | 0.244 | 0.322 | −0.304 | 0.043 | 0.519 |
| | Sig. | | 0.131 | 0.000 | 0.000 | 0.000 | 0.000 | 0.500 | 0.000 |
| Women Weakness | $r_s$ | 0.144 | 1.000 | 0.013 | −0.491 | −0.389 | 0.208 | 0.551 | −0.007 |
| | Sig. | 0.022 | | 0.834 | 0.000 | 0.000 | 0.001 | 0.000 | 0.913 |
| Women Dominance | $r_s$ | −0.249 | 0.000 | 1.000 | 0.247 | 0.137 | 0.552 | 0.196 | −0.172 |
| | Sig. | 0.000 | 0.994 | | 0.000 | 0.032 | 0.000 | 0.002 | 0.007 |
| Women Agency | $r_s$ | 0.215 | −0.499 | 0.246 | 1.000 | 0.714 | −0.030 | −0.380 | 0.441 |
| | Sig. | 0.001 | 0.000 | 0.000 | | 0.000 | 0.643 | 0.000 | 0.000 |
| Men Communality | $r_s$ | 0.325 | −0.374 | 0.131 | 0.707 | 1.000 | 0.146 | −0.554 | 0.269 |
| | Sig. | 0.000 | 0.000 | 0.039 | 0.000 | | 0.022 | 0.000 | 0.000 |
| Men Weakness | $r_s$ | −0.329 | 0.181 | 0.554 | −0.025 | 0.130 | 1.000 | 0.064 | −0.440 |
| | Sig. | 0.000 | 0.004 | 0.000 | 0.700 | 0.041 | | 0.319 | 0.000 |
| Men Dominance | $r_s$ | 0.081 | 0.554 | 0.185 | −0.385 | −0.537 | 0.042 | 1.000 | 0.126 |
| | Sig. | 0.205 | 0.000 | 0.004 | 0.000 | 0.000 | 0.514 | | 0.049 |
| Men Agency | $r_s$ | 0.543 | 0.015 | −0.184 | 0.427 | 0.279 | −0.456 | 0.145 | 1.000 |
| | Sig. | 0.000 | 0.811 | 0.004 | 0.000 | 0.000 | 0.000 | 0.022 | |

Partial correlation by Spearman's rank coefficient showed that demographic variables related to all correlations (see upper-right partial correlation matrix, Table 4).

To obtain a clearer profile of prospective teachers' beliefs about gender in society, we further examined the mean values of the four categories (i.e., communality, weakness, dominance, and agency) by using line graphs to form graphical indices. By examining the means, it became obvious which characteristics were considered desirable or undesirable for women and men in society. In this way, we can specify beliefs about gender and more precisely outline the profiles of prospective teachers.

The graphical comparison of the mean values of the men and women in the sample showed that communal characteristics were considered desirable for women and men, with means between 4.00 and 7.00 (Figures 1 and 2). In particular, the traits "sensitive" and "sympathetic" were desirable in women, as the line graphs of female and male prospective teachers were very close for these traits (Figure 1). The mean values showed that female prospective teachers believed more strongly than male prospective teachers in that communality is desirable for women in society. The men's mean values, similar to those of the women, were almost entirely in the range between 5.00 and 7.00. From the men's point of view, communal characteristics are, thus, desirable traits for women as well. Although the tendency of men's and women's evaluations of the communal traits were similar, the calculation of the effect size showed that men and women differed in their ratings of three traits: Thus, Cohen's d showed medium effects for the traits "helpful to others" (|d| = 0.585), "understanding of others" (|d| = 0.522), and soft-hearted (|d| = 0.508).

Concerning communal traits in men, the mean values were close with Cohen's d, finding no effects that might indicate a meaningful difference (Figure 2). However, Cohen's d proved that male prospective teachers evaluated the traits "warm" and "compassionate" differently for men than for women (warm |d| = 0.601; compassionate |d| = 0.561). In addition, female prospective teachers showed differences in their evaluations of these characteristics in men and women ("warm" |d| = 0.757; "compassionate" |d| = 0.744).

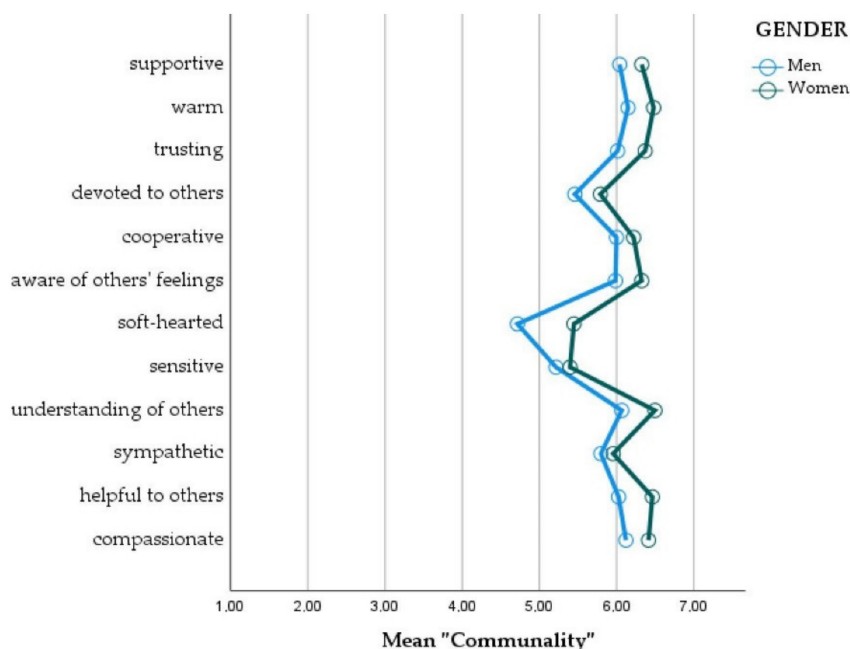

**Figure 1.** Prospective teachers' beliefs about communal gender stereotypes for women in society (1.00 = not at all desirable, 7.00 = very desirable).

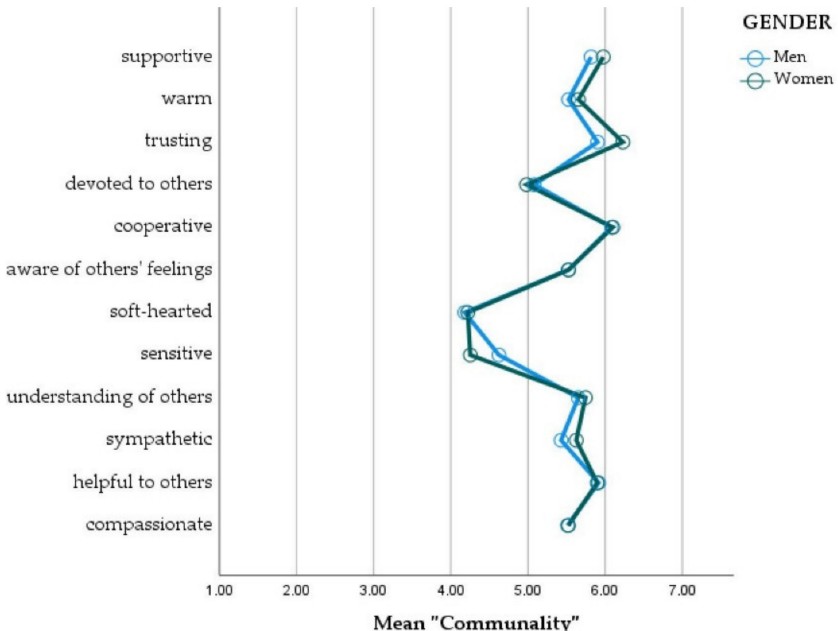

**Figure 2.** Prospective teachers' beliefs about communal gender stereotypes for men in society (1.00 = not at all desirable, 7.00 = very desirable).

Weak traits were less desirable for women (Figure 3) and men (Figure 4) with means predominantly between 1.00 and 4.00. However, female prospective teachers had higher mean values and rated weak traits as more desirable for women, compared to male prospective teachers. Most desirable traits in women were "approval seeking" and "worried" with values close to 4.00 and 5.00.

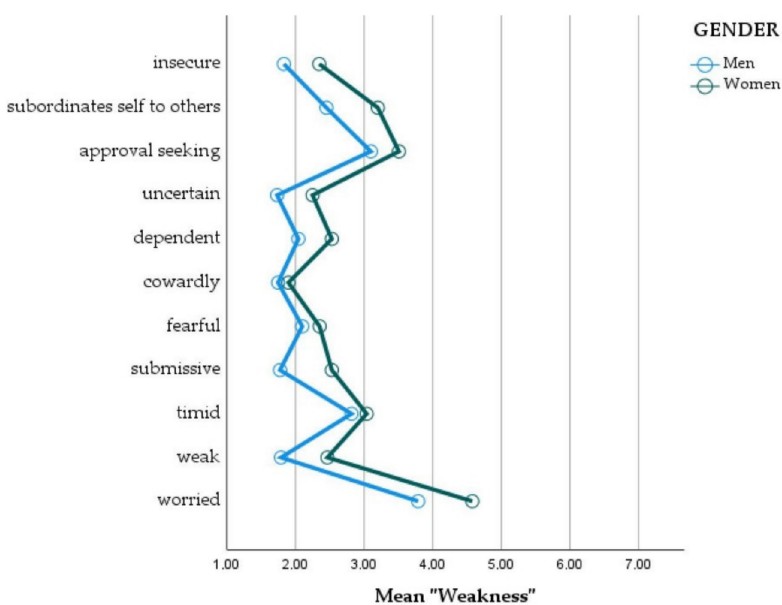

**Figure 3.** Prospective teachers' beliefs about weak gender stereotypes for women in society (1.00 = not at all desirable, 7.00 = very desirable).

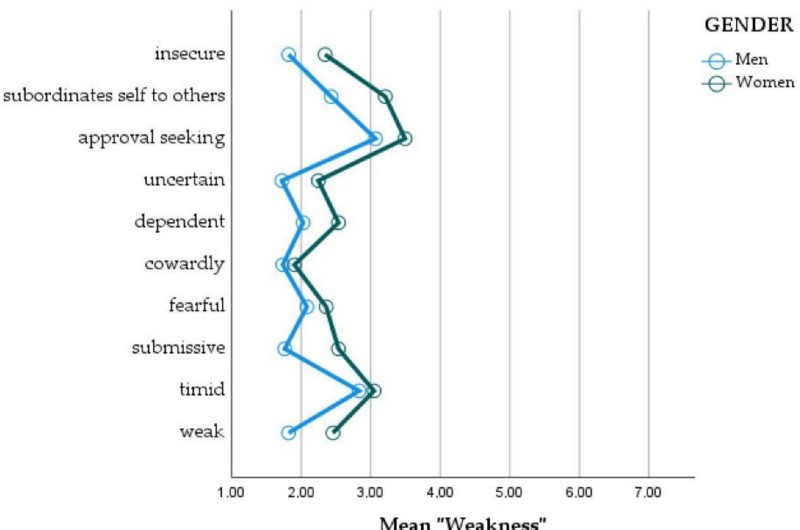

**Figure 4.** Prospective teachers' beliefs about weak gender stereotypes for men in society (1.00 = not at all desirable, 7.00 = very desirable).

In contrast, male prospective teachers had lower mean values when it came to the desirability of weak traits for men in society (Figure 4). Especially, traits such as "subordinates self to others", "submissive", and "weak" were not desirable in men. Although there are gaps visible in the line graph for men, the values reflected the same trend for weak traits for men and women in society. Cohen's d showed no significant effects regarding the desirability of weak traits, either within, or between the groups.

Dominant traits were less desirable for women (Figure 5) and men (Figure 6) with mean values mainly between 1.00 and 4.00. Female prospective teachers especially rejected the dominant traits "boastful", "aggressive", "forceful", "dominant", and "demanding" in women. For some of the traits, male prospective teachers showed even lower mean values than female prospective teachers, especially rejecting traits such as "dictatorial", "bossy", and "controlling" in women. Nevertheless, the values of men and women were extremely close to each other, thus neither strong nor moderate effect sizes can be reported.

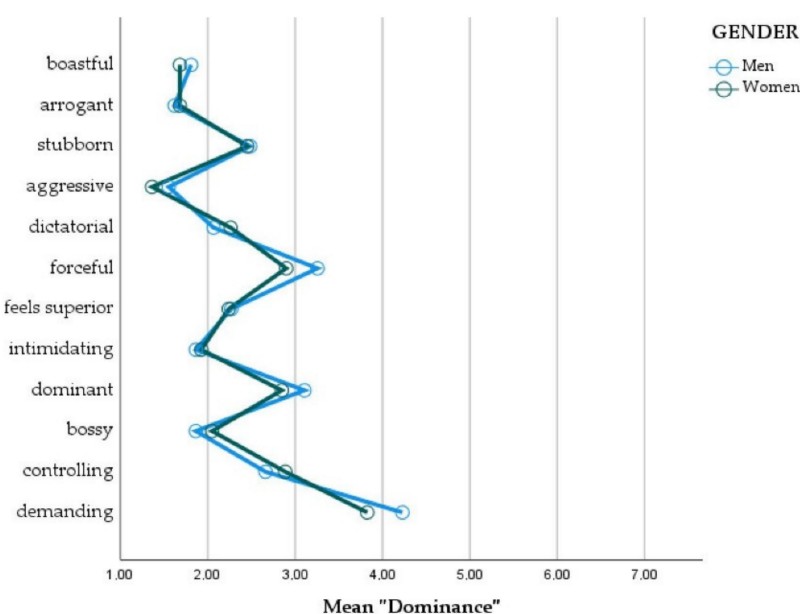

**Figure 5.** Prospective teachers' beliefs about dominant gender stereotypes for women in society (1.00 = not at all desirable, 7.00 = very desirable).

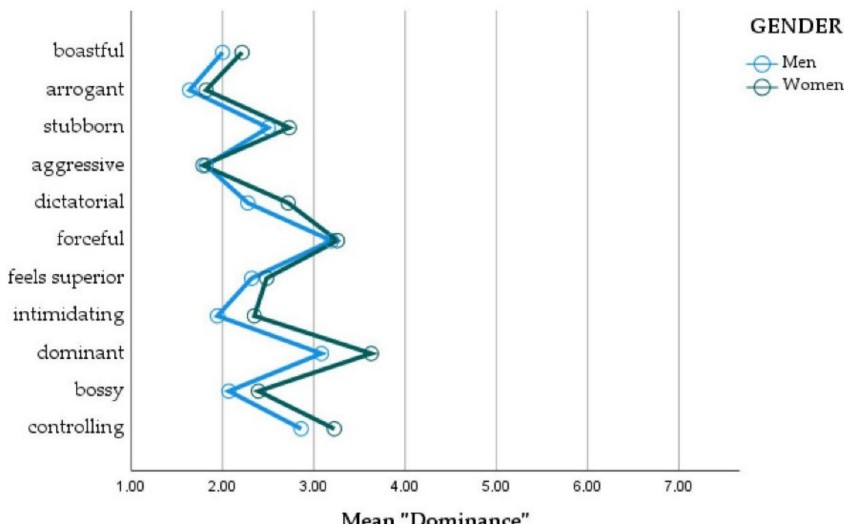

**Figure 6.** Prospective teachers' beliefs about dominant gender stereotypes for men in society (1.00 = not at all desirable, 7.00 = very desirable).

In contrast, female prospective teachers considered dominant traits slightly more desirable for men in society, having higher mean values than male prospective teachers (Figure 6). The mean values ranged from 1.00 to 4.00, and the tendency could be established that dominant traits were also rather undesirable in men.

Agentic traits were desirable for women (Figure 7) and highly desirable for men (Figure 8). With one exception ("competitive"), the mean values for women were between 4.00 and 7.00, similar to the mean values for men. Female prospective teachers considered the traits "independent", "has leadership abilities", and "competitive" as the least desirable in women (Figure 7). Apart from the trait "competitive" ($|d| = -0.568$), agency for women was consistently seen as desirable from the perspective of female prospective teachers.

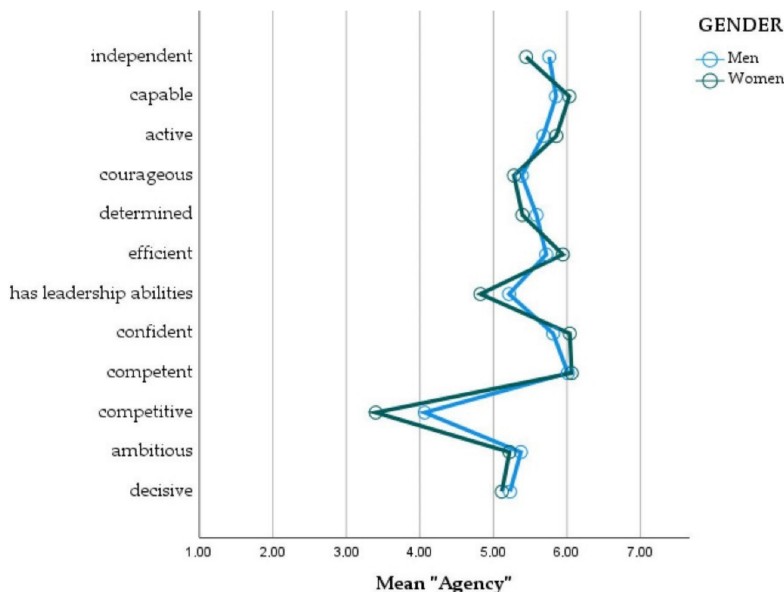

**Figure 7.** Prospective teachers' beliefs about agentic gender stereotypes for women in society (1.00 = not at all desirable, 7.00 = very desirable).

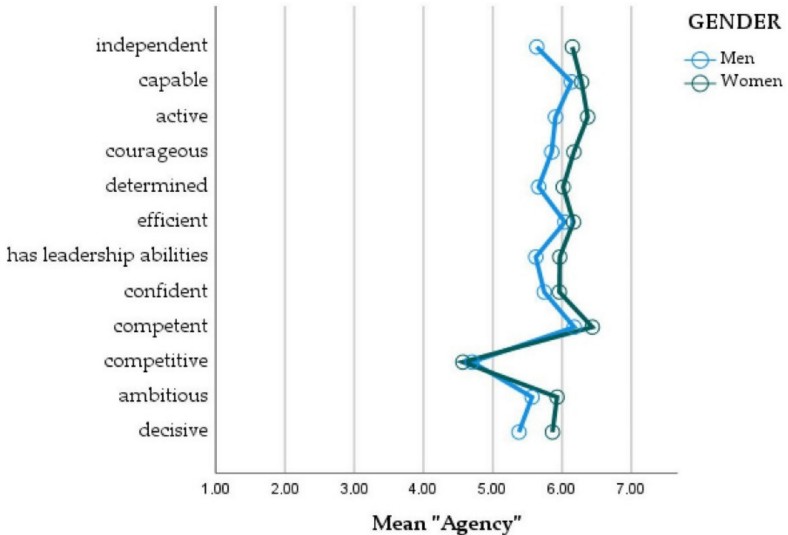

**Figure 8.** Prospective teachers' beliefs about agentic gender stereotypes for men in society (1.00 = not at all desirable, 7.00 = very desirable).

Compared to male prospective teachers, female prospective teachers considered agentic traits more desirable in men; they considered the traits "independent", "active", and "decisive" to be particularly desirable in men (Figure 8). Female prospective teachers evaluated agentic traits differently for women and men in terms of desirability: Thus, Cohen's d showed medium effects for the trait "independent" (|d| = −0.582) and strong effects for the trait "competitive" (|d| = −0.818) when comparing the desirability of these traits for women and men.

### 3.3. Teachers' Beliefs about Math

The second research question addressed the beliefs prospective teachers have about math and about female and male students' competencies in math. Central to the analysis was the gender of the prospective teachers and the subject on which they have chosen to focus on, that is, whether they aspire to become mathematics teachers or prospective

teachers of another subject. To determine the extent to which female and male prospective math and non-math teachers differ significantly in terms of their beliefs about math, the MANCOVA was performed with the sample divided by gender, using subject as a factor, and controlling for area of growing up (used as second factor) and age (used as a covariate).

Concerning prospective teachers' beliefs about math, the MANCOVA found statistically significant differences between the subjects on "environment", "career", and "attribution of success" for the sample of female prospective teachers (Table 5). There was homogeneity of covariances, as assessed by Box's M test ($p > 0.001$). Homogeneity of variances was asserted using Levene's Test, which showed that equal variances can be assumed for all dependent variables ($p > 0.05$), except for "career" ($p = 0.016$). Therefore, we calculated the Welch ANOVA, thus providing a significant result and confirming a statistically significant difference for "career" for the sample's proportion of women, $F(1, 92.13) = 19.70$, $p < 0.001$.

**Table 5.** Female prospective teachers' beliefs about math related to subject, controlling for growing up area, and age (from left to right: means, MANCOVA, partial eta squared).

| | Subject (S) [a] | | Growing Up Area (GA) [a] | | S | GA | Age [b] | Partial $\eta^2$ [c] |
|---|---|---|---|---|---|---|---|---|
| | 1 | 2 | 1 | 2 | F, p df | F, p df | F, p df | |
| Environment | 2.70 | 3.04 | 2.56 | 2.97 | 5.47 ** | 9.47 ** | 1.03 ns | |
| | 115 | 41 | 70 | 86 | 1/151 | 1/151 | 1/151 | 11% |
| Career | 3.11 | 3.70 | 3.17 | 3.35 | 16.00 *** | 1.92 ns | 2.99 ns | |
| | 115 | 41 | 70 | 86 | 1/151 | 1/151 | 1/151 | 13% |
| Competence | 1.82 | 1.78 | 1.63 | 1.96 | 0.03 ns | 3.19 ns | 1.13 ns | |
| | 115 | 41 | 70 | 86 | 1/151 | 1/151 | 1/151 | 3% |
| Attribution of Success | 1.94 | 2.27 | 1.86 | 2.16 | 5.75 ** | 10.05 ** | 0.27 ns | |
| | 115 | 41 | 70 | 86 | 1/151 | 1/151 | 1/151 | 13% |
| Results MANCOVA, Pillai's Trace | | | | | 5.22 *** | 3.86 ** | 0.78 ns | |
| | | | | | 4/604 | 4/604 | 4/604 | |

Scale: 1 = disagree strongly–5 = agree strongly; [a] mean values and number of cases of the factor level (S): 1 = Math; 2 = No Math; (GA): 1 = Urban; 2 = Rural; [b] Covariate; [c] Effect size showing percentage of variance associated with an effect; ** $p < 0.01$; *** $p < 0.001$; ns = not significant.

The MANCOVA showed no significance when controlling for age, but after adjusting for the area of growing up, there were significant findings ("environment" and "attribution of success"). For men, MANCOVA found no statistically significant differences between the subject on the combined dependent variables, $F(4, 57) = 0.334$, $p = 0.854$, partial $\eta^2 = 0.023$. Controlling for the area of growing up and for age, there were no statistically significant differences between either the area of growing up ($F(4, 57) = 1.306$, $p = 0.279$, partial $\eta^2 = 0.084$) or between the age of the men in the sample on the combined dependent variables ($F(4, 57) = 1.740$, $p = 0.154$, partial $\eta^2 = 0.109$). Due to the non-significant MANCOVA result, data for men were not presented in a table.

The analysis of the mean values showed that women have stronger beliefs about math. This relates to the dimension "environment", which measured prospective teachers' beliefs about how parents and peers perceive male students in math (e.g., the environment perceives boys as more competent in math). Female prospective teachers who do not teach math agreed more with these statements than female prospective math teachers.

The dimension "career" examined prospective teachers' beliefs about the gender appropriateness of students' career choices (e.g., boys are more interested in careers that require mathematical abilities). It showed that female prospective, non-math teachers agreed more strongly with these statements.

The dimension "attribution of success", which showed statistically significant differences in the MANCOVA, pointed out that female prospective teachers who do not teach math also agreed more likely with statements about the reasons for student achievement

in math (e.g., boys are more successful in math because they receive more support from teachers and parents).

However, as MANCOVA results suggested, subject and the place of growing up played an important role in prospective teachers' beliefs, as it had an influence on the dimensions "environment" and "attribution of success". When controlling for the area of growing up, the dimension "competence" became significant while "career" was no longer significant, which meant that the factor did not adjust this association. Overall, beliefs about math depended on gender, subject, and the area of growing up.

### 3.4. Relations between Teachers' General Beliefs about Gender and Their Beliefs about Math

Finally, the third research question answered how prospective teachers' general beliefs about gender are related to beliefs about math by performing a partial correlation with variables of the scales "pre-/proscription women", "pre-/proscription men", and "teachers' gender stereotypes toward mathematics". According to the Shapiro–Wilk test, there was no normal distribution ($p < 0.05$). Therefore, the nonparametric Spearman's rank-order correlation was used, controlling for demographic variables (i.e., age and area of growing up).

Concerning prospective teachers' beliefs about math, the results of the second research question already showed that beliefs about math depend on gender, subject, and the place of growing up. Due to the fact that women who do not aspire to become math teachers, as well as women who grew up in rural areas, have stronger beliefs than men do about math, the partial correlation focused only on female prospective teachers (Table 6).

Looking at the results of the partial correlations, female prospective teachers, in terms of their beliefs about gender in society, showed significant correlations in the variables that emphasized the stereotypical dynamics of communal, weak, dominant, and agentic characteristics.

Moderately significant positive correlations existed between dominant traits in men and weak traits in women ($r_s = 0.591$), and between agentic traits in men and communal traits in women ($r_s = 0.564$). The strongest positive correlation was between communal traits in men and agentic traits in women ($r_s = 0.714$).

There were also moderate correlations within gender, for example, for agentic and weak traits in men ($r_s = 0.505$), and dominant and communal traits in men ($r_s = -0.572$). Regarding desirable traits in women, there was a moderately negative correlation between agency and weakness ($r_s = -0.571$).

Looking at the correlations between the variables reflecting the desirable traits of men and women, and those reflecting gender-related math stereotypes, there were seven weak but significant correlations. Two of these referred to weakness in women (weakness and career, $r_s = 0.168$; weakness and attribution of success, $r_s = 0.224$), and the remaining five correlations involved communality and dominance in men. Thus, there were significant negative correlations between communality and environment ($r_s = -0.157$), communality and career ($r_s = -0.160$), and communality and attribution of success ($r_s = -0.175$). In addition, there were positive correlations between dominance and career ($r_s = 0.184$), and dominance and attribution of success ($r_s = 0.207$).

Partial correlation by Spearman's rank coefficient showed that demographic variables related to all correlations between beliefs about gender and gender-related math beliefs (see upper-right partial correlation matrix, Table 6).

**Table 6.** Partial correlation matrix for female prospective teachers (using Spearman's rank-order correlations): The lower-left correlations are unadjusted, and the upper-right correlations are partial correlations adjusted for age and area of growing up.

| | | Women Communality | Women Weakness | Women Dominance | Women Agency | Men Communality | Men Weakness | Men Dominance | Men Agency | Environment | Career | Competence | Attribution of Success |
|---|---|---|---|---|---|---|---|---|---|---|---|---|---|
| Women Communality | $r_s$ | 1.000 | 0.132 | −0.277 | 0.187 | 0.276 | −0.391 | 0.052 | 0.559 | 0.082 | 0.129 | −0.021 | 0.042 |
| | Sig. | | 0.078 | 0.000 | 0.012 | 0.000 | 0.000 | 0.490 | 0.000 | 0.286 | 0.100 | 0.787 | 0.597 |
| Women Weakness | $r_s$ | 0.120 | 1.000 | −0.157 | −0.570 | −0.470 | 0.048 | 0.593 | 0.048 | 0.125 | 0.161 | 0.092 | 0.204 |
| | Sig. | 0.108 | | 0.036 | 0.000 | 0.000 | 0.526 | 0.000 | 0.527 | 0.102 | 0.040 | 0.228 | 0.009 |
| Women Dominance | $r_s$ | −0.275 | −0.158 | 1.000 | 0.334 | 0.204 | 0.573 | 0.107 | −0.210 | −0.113 | −0.113 | 0.049 | 0.038 |
| | Sig. | 0.000 | 0.034 | | 0.000 | 0.007 | 0.000 | 0.157 | 0.005 | 0.141 | 0.151 | 0.525 | 0.635 |
| Women Agency | $r_s$ | 0.194 | −0.571 | 0.332 | 1.000 | 0.718 | 0.026 | −0.390 | 0.348 | −0.128 | −0.115 | −0.061 | −0.092 |
| | Sig. | 0.009 | 0.000 | 0.000 | | 0.000 | 0.729 | 0.000 | 0.000 | 0.095 | 0.145 | 0.428 | 0.245 |
| Men Communality | $r_s$ | 0.279 | −0.476 | 0.207 | 0.714 | 1.000 | 0.161 | −0.573 | 0.213 | −0.144 | −0.155 | −0.105 | −0.159 |
| | Sig. | 0.000 | 0.000 | 0.006 | 0.000 | | 0.033 | 0.000 | 0.004 | 0.061 | 0.049 | 0.173 | 0.045 |
| Men Weakness | $r_s$ | −0.398 | 0.060 | 0.569 | 0.017 | 0.151 | 1.000 | −0.026 | −0.498 | −0.159 | −0.128 | 0.051 | 0.116 |
| | Sig. | 0.000 | 0.425 | 0.000 | 0.821 | 0.044 | | 0.729 | 0.000 | 0.038 | 0.106 | 0.510 | 0.144 |
| Men Dominance | $r_s$ | 0.053 | 0.591 | 0.106 | −0.388 | −0.572 | −0.026 | 1.000 | 0.202 | 0.057 | 0.184 | 0.084 | 0.209 |
| | Sig. | 0.486 | 0.000 | 0.162 | 0.000 | 0.000 | 0.730 | | 0.007 | 0.463 | 0.020 | 0.275 | 0.008 |
| Men Agency | $r_s$ | 0.564 | 0.030 | −0.209 | 0.356 | 0.213 | −0.505 | 0.201 | 1.000 | −0.027 | 0.121 | 0.021 | 0.024 |
| | Sig. | 0.000 | 0.688 | 0.005 | 0.000 | 0.004 | 0.000 | 0.007 | | 0.730 | 0.126 | 0.787 | 0.762 |
| Environment | $r_s$ | 0.065 | 0.143 | −0.114 | −0.137 | −0.157 | −0.137 | 0.058 | −0.050 | 1.000 | 0.493 | 0.220 | 0.343 |
| | Sig. | 0.397 | 0.059 | 0.135 | 0.072 | 0.039 | 0.073 | 0.448 | 0.516 | | 0.000 | 0.004 | 0.000 |
| Career | $r_s$ | 0.121 | 0.168 | −0.113 | −0.119 | −0.160 | −0.119 | 0.184 | 0.107 | 0.497 | 1.000 | 0.375 | 0.383 |
| | Sig. | 0.121 | 0.031 | 0.147 | 0.128 | 0.041 | 0.131 | 0.019 | 0.174 | 0.000 | | 0.000 | 0.000 |
| Competence | $r_s$ | −0.042 | 0.111 | 0.048 | −0.076 | −0.114 | 0.073 | 0.081 | −0.021 | 0.244 | 0.380 | 1.000 | 0.337 |
| | Sig. | 0.584 | 0.142 | 0.527 | 0.316 | 0.136 | 0.338 | 0.289 | 0.785 | 0.001 | 0.000 | | 0.000 |
| Attribution of Success | $r_s$ | 0.022 | 0.224 | 0.033 | −0.103 | −0.175 | 0.136 | 0.207 | −0.006 | 0.367 | 0.388 | 0.361 | 1.000 |
| | Sig. | 0.783 | 0.004 | 0.678 | 0.190 | 0.025 | 0.085 | 0.008 | 0.936 | 0.000 | 0.000 | 0.000 | |

## 4. Discussion

Regarding the first research question (i.e., what general beliefs prospective teachers have about gender), the following can be concluded. The evaluation of the self-descriptions of prospective teachers showed that, although men also attributed communal traits to themselves, women had stronger beliefs about their own gender regarding communality. Communality seemed to be a key factor for female prospective teachers in defining themselves as women. This finding was consistent with international research showing that women not only are generally viewed as more communal but that they also attributed more communality to themselves [66,72].

Essentialist beliefs about gender could not be verified. However, the gender role ideology beliefs of female and male prospective teachers indicated an orientation toward an egalitarian division of labor. The traditional distribution of roles was replaced by equal responsibility for family and income, at least in theory. It is likely that the fact that the respondents aspired to the teaching profession and studied at Universities of Teacher Education influenced this result. Moreover, in 2020, 60% of the Swiss population voted in favor of the introduction of two-week paid paternity leave for fathers. As studies showed that culture and society play an important role in how people perceive themselves and others [68,73,74], a societal shift in thinking could have also affected the respondents' beliefs about gender and gender roles.

The findings on gender beliefs in society reflected the research findings on pre- and prospective gender stereotypes [67,75,76]: Communality and weakness are traits considered desirable for women, but agency and dominance are for men. Compared to male prospective teachers, female prospective teachers viewed weak traits in women as more desirable. As social role theory describes, women and men learn early which gender roles and gender-typical behavior are expected of them [36]. Therefore, to meet society's expectations, women may be more willing to accept less advantageous characteristics, such as being weak. For women in male-dominated STEM occupations, for example, research has proven that they have a "stigma consciousness", according to which they adjust their appearance and/or behavior to avoid negative judgment and stereotypes [77]. It is known from research that, when women violate proscriptive stereotypes by being non-role conforming (e.g., dominant), they were less liked and, therefore, less likely to be hired, even though they were viewed as competent [75]. However, men are also aware of proscriptive gender stereotypes [78]. Male prospective teachers rejected weakness in men more than female prospective teachers. Here, too, socialization played an important role, as men have learned that weakness in society tends to be equated with femininity and a low status in gender hierarchy and, therefore, needs to be avoided [28,75]. In contrast, communality and agency are desirable for women and men. However, the data suggested that those communal traits, which are associated more with femininity, are considered less desirable for men in society. Gradual differences can also be observed for agency. Agentic traits, which are more associated with masculinity, are considered less desirable for women in society [72].

The second research question aimed to determine what beliefs prospective teachers have about math and how they perceive students' competencies in math. Female prospective teachers who do not teach math and who share rural socialization experiences agreed more with boys being perceived as more competent in math by their parents and peers ("environment"), indicating that they stereotype math as a male domain [69]. In addition, female prospective non-math teachers agreed that boys benefit more from the additional effort of teachers and parents than girls ("attribution of success"), suggesting that boys were perceived to be more talented in math [52,69]. These results confirmed findings from studies that, even among teachers, math has a strong male connotation [43–45]. Similarly, the results were consistent with research, showing that teachers share math–male gender stereotypes, ascribing more talent to male than to female students [12,52,53,57,79].

Although an influence of age could not be demonstrated for either gender or mathematics beliefs, the area of growing up proved significant. Rural socialization experiences

influenced female prospective teachers' beliefs about math. Growing up in rural areas of Switzerland, where women had to wait a long time for full civil rights, may have led to internalizing more regressive beliefs about gender, which have persisted there longer than in Switzerland's urban areas.

Furthermore, female prospective teachers who do not teach math agreed more with boys being better suited for STEM careers because of their greater mathematical abilities and interest ("career"). The results could be attributed to the observation of boys' and girls' career choice behaviors [80]. However, because female prospective teachers, in terms of their own career choices, have avoided math and math-related careers, there was much to suggest that the results reflected the actual beliefs of female prospective teachers.

The third research question connected the first two questions by examining how prospective teachers' general beliefs about gender relate to beliefs about mathematics and boys' and girls' competencies in mathematics. The results suggested that female prospective teachers who consider dominance in men and weakness in women to be desirable also believe that STEM careers are more suitable for boys ("career"), and attributed more mathematical talent to boys than girls ("attribution of success"). Negative correlations existed between gender-related beliefs about math and beliefs about communality among men. It can be concluded that beliefs about communal traits being considered less desirable in men are accompanied by beliefs about math being a male domain ("environment"), beliefs about boys being better suited for STEM careers ("career"), and beliefs about boys being more talented in math than girls ("attribution of success"). According to this, communality was contrary to the masculine image of math, which was only conclusive because communality, in particular, is a key factor when it comes to identifying as a woman [66,72]. Complementing Aslan's study [81] where teachers most often described boys as strong and rational, and girls as sensitive, fragile, and emotional, it can be concluded that in stereotypes where the weak girl and the non-communal, dominant boy are prevalent, math stereotypes about boys and girls can most likely be expected.

## 5. Conclusions

Overall, our study brought to light several important findings and, therefore, enriched the scientific discourse. Research during the past 20 years has shown that teachers' gender beliefs about mathematics were significantly in favor of boys. In addition, our study provided the first empirical evidence of the relationship between general gender stereotypes and math stereotypes. Therefore, we created a differentiated profile of prospective teachers by examining their beliefs about their self-image, their images of men and women in society, their essentialist and gender-role ideology beliefs, and their math stereotypes. Then, we linked prospective teachers' beliefs about gender (based on 48 characteristics) to their beliefs about mathematics and about girls' and boys' competencies in math. The extensive analysis provided knowledge about prospective teachers, which is particularly important for teacher education.

Although, in international comparison, Switzerland is one of the 10 countries with the most advanced implementation of gender equality [40], stereotypical ideas about gender still prevail among prospective teachers. Accordingly, communality was important for female identification, whereas dominant traits in women (i.e., those partly needed for STEM careers) were viewed as undesirable. In contrast, the avoidance of weakness was a central concern for men.

Another important finding was that socialization experiences contributed significantly to differences between gender-related beliefs about math. Compared to female prospective math teachers who grew up in urban areas, women who grew up in rural areas were more likely to stereotype math and their students. Moreover, women without a math background tended to have math stereotypes in terms of the subject's image as male-dominated, career choices, and beliefs that boys are more talented in math. This is important to note because biases, regardless of what subject the prospective teacher will be teaching, contribute to

students internalizing them. This has a negative impact on girls' and young women's math ability self-concept and future career choices, as they are less likely to choose STEM careers.

Most importantly, our study provided empirical evidence that stereotypical beliefs about math relate to stereotypical beliefs about gender. The connection of general gender stereotypes and gender-related math stereotypes seemed to be particularly problematic for prospective teachers, who stereotypically associated femininity with weakness and masculinity with non-communal dominance but perceived math as a male domain, considering boys more talented and suitable for STEM careers than girls. "Gender and equality" are explicitly defined as an educational goal to be taught in Swiss schools [82]. Ideally, prospective teachers should reflect this educational goal in their personal attitudes. However, our study showed that this was not the case for all prospective teachers. Teacher education is challenged to point out to prospective teachers the effects of unreflective and discriminatory gender bias on their students. Raising (prospective) teachers' awareness is an important precondition for teaching and for achieving gender equality in education.

For future studies, it is essential to link quantitative data with qualitative data to gain a comprehensive understanding of prospective teachers' beliefs about gender and mathematics. In our data collection survey, we additionally asked three open-ended questions about prospective teachers' beliefs about masculinity and femininity, therefore, we plan to conduct content analysis on these in a next step and combine the results with the findings presented in this paper.

In addition, further research is needed on how general gender stereotypes affect (prospective) teachers' beliefs and, more importantly, how they affect students' beliefs about learning, achievement, and career choice. Critical examination and conscious questioning of one's own stereotypical beliefs will lead, in the long term, to girls and boys having equal opportunities to shape their life plans free from restrictive gender stereotypes. As research has shown, gender attitudes, particularly the role of women, changed significantly during the past 70 years [83]. Therefore, the change toward an equal world is not utopian and begins in small ways. Hopefully, our study's results will serve as a basis for improving gender equality in education and, in doing so, also improving gender equality in society for future generations.

## 6. Limitations

One aspect of the study that can be criticized is that dichotomous gender relations were reproduced in the survey by questions and answers referring to binary categories of gender and gender roles. However, the present study was conducted as part of a cross-cultural study. Its goal was to examine and compare beliefs about gender and gender stereotypes in more than 70 countries. Therefore, the survey had to reflect the regressive and progressive beliefs of participants from countries with different levels of gender equality. Nevertheless, care was taken to use gender-sensitive language in the survey's German translation.

Moreover, prospective teachers participated voluntarily in the survey, that is, they might have a potential interest in the topic. Therefore, caution should be exercised when generalizing the results. In addition, it should be noted that the study represented only the German-speaking part of Switzerland.

Another aspect that should be critically emphasized concerns the scales used in the survey. In interpreting the results, we assumed that the participants correctly understood the questions measuring prospective teachers' beliefs and answered according to their own beliefs. However, we cannot rule out that participants based agreement with some statements on their own beliefs or on observations of real facts. As the reliability of the scales suggested that the instruments were valid, we assumed that the scales worked correctly, and the data indeed reflected the beliefs of prospective teachers.

Additionally and notably, regarding the interpretation of the applied partial correlations, the results were not entirely clear: The interpretation can be made in two directions. Moreover, the results did not allow us to draw conclusions about the causality of the relationships. We interpreted the partial correlations as far as the state of research allowed,

and to address this limitation, we examined results using graphical indices to compare the mean values of men and women in more detail.

Finally, the presentation of the results focused strongly on the significant differences found between male and female prospective teachers, as well as between prospective math and non-math teachers. However, despite significant results, the mean values were often close and reflected respondents' similar tendencies and beliefs. Due to the contrasting nature of the results, stereotypical beliefs were emphasized more. Nevertheless, it must be noted that the respondents never showed complete agreement with stereotypical statements.

**Author Contributions:** Conceptualization, J.L. and E.M.; methodology, J.L., E.M., D.B. (Dorothee Brovelli) and D.B. (Deborah Bernhard); validation, J.L., E.M., D.B. (Dorothee Brovelli) and D.B. (Deborah Bernhard); formal analysis, J.L.; investigation, J.L.; resources, J.L. and E.M.; data curation, J.L. and D.B. (Deborah Bernhard); writing—original draft preparation, J.L.; writing—review and editing, J.L., E.M., D.B. (Dorothee Brovelli) and D.B. (Deborah Bernhard); visualization, J.L.; project administration, J.L. All authors have read and agreed to the published version of the manuscript.

**Funding:** This research received no external funding.

**Institutional Review Board Statement:** The Ethics Board for Research Projects at the Institute of Psychology, University of Gdańsk (approval code 11/2018, date of approval 13 November 2018), reviewed and approved the cross-cultural comparative study.

**Informed Consent Statement:** Informed consent was obtained from all subjects involved in the study.

**Data Availability Statement:** Because the data are currently analyzed as part of a doctoral thesis, the dataset will be available from the corresponding author on request after completion of the dissertation.

**Conflicts of Interest:** The authors declare no conflict of interest.

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
