# Peer review of "Toward Gender Equality in Education—Teachers’ Beliefs about Gender and Math"

_education, doi:10.3390/educsci12060373_

Round 1

Reviewer 1 Report

Review of “Towards Gender Equality in Education—Teachers’ Beliefs about Gender and Math”

7 April 2022

This manuscript describes an analysis of survey data on gender beliefs and gendered beliefs about math. The goal of the study is to determine prospective teachers’ beliefs about gender/gender roles, and to connect them to the teachers’ beliefs about gender and math.

I find the topic to be very interesting, and the research questions valid and of value. This could be a useful study if it is improved. As it stands, there are too many issues with the paper to recommend publication. I offer comments to the authors to guide their revisions.

I found the language to be a bit difficult to understand (English is my first language); shorter sentences would make the reading easier.

In the literature review/introduction, there is nothing about age, so I was not sure why the authors chose to include this as a covariate. It is not mentioned in the discussion, either.

The comments in the discussion about rural/urban backgrounds would be better in the introduction to give a rationale for why this was studied. I find the argument they made very convincing about why it was important to look at region.

The literature includes a mix of studies looking at prospective or in-service teachers. Where comparisons are being made to studies of in-service teachers, a short note would be helpful to remind readers of this difference.

I approve of trying to include something about respondents’ belief about gender role/ideology, but I do not think that the question asked really gets at what the authors are hoping it would. I think this analysis should be removed.

The MANCOVA tables were hard to interpret; it took a few tries to connect up the text with the table. Having more information in the headers would be helpful.

I was wondering why there was no MANCOVA analysis on the gender beliefs for men and women in society. Also, I think the men’s graphical data should be pulled out of the appendix and put in the manuscript with the women’s, to make it easy to compare.

One of the biggest concerns I have with the manuscript is the focus solely on gaps while ignoring the actual values of the data. For instance, line 615 notes the difference between math and non-math teachers, but both values are thankfully low, around 2 on the 1-5 scale.

A related issue is that several of the claims are overstated. Line 440 talks about the difference between Figure 2 and Figure A2, but the differences are very small. Figures 1 and 2 have very small differences, and 3 and 4 are pretty much the same. For the men, only A2 has much of the difference. Again, a discussion of the values is important, to go alongside the analysis of the gaps.

I did not see any data presented for the statement in line 500; perhaps I missed it?

The bolding on the large correlation tables did not match with what was explained in the text, so I wasn’t sure what the bold meant.

Overall, this has merit, but needs significant work. I hope the authors consider reworking some of the analysis and making the suggested changes.

Reviewer 2 Report

I suggest that the following be unpacked more: "

Consequently, girls do not generally perform significantly worse than boys. Moreover, gender differences in terms of performance in averages are very small. Girls in the 8th grade perform exceptionally well in algebra and  geometry compared to boys. Beliefs whereas competencies in math must inevitably be linked to gender are thus invalidated"

Reviewer 3 Report

This paper deals with teachers’ beliefs about gender and their gender-related image of mathematics. Specifically, this is the first work providing empirical evidence of the relationship between general gender stereotypes and math stereotypes among prospective teachers. The authors perform statistical analysis on a sample of 275 male and female prospective teachers using partial correlation and MANCOVA analysis. The findings in terms of gender beliefs are consistent with the literature: for example, among the prescriptive gender stereotypes, communality and weakness are traits considered desirable for women, while agency and dominance are for men. Regarding the proscriptive gender stereotypes, the results show that communality is less desirable in men and agentic traits in women. In terms of beliefs about mathematics, women who grew up in rural areas and who intended to teach subjects other than mathematics perceived boys as more competent than girls in math. As for the relationship between gender beliefs and gender-related image of math, the authors found among others that female prospective teachers who considered desirable dominance in men and weakness in women also believed that STEM careers are more suitable for boys and that boys are more talented in math than girls. 

This paper addresses a truly relevant issue in education. The introduction efficiently motivates the investigation and places the work in the existing knowledge. The bibliography is comprehensive, up to date and appropriate. The methodology is adequate, and the computations in the findings appear to be correct. Tables and graphs are adequate and easy to interpret. Conclusions are consistent with the evidence and arguments presented. In general, the paper is well-structured and uses quite good language and vocabulary. From my point of view, this study is worth publishing.  Here are some minor typos that the authors may wish to correct: 

(Line 92) It says: “...group respectively community”. I do not understand this sentence. Is there a comma missing? You should clarify this. 

(Line 243) “in another subject than math” should be “in a subject other than math” 

(Line 260) “...how the participating women and men perceive themselves respectively which...”.  I think a comma is missing between ‘respectively’ and ‘which’. 

(Line 263) “...and men respectively which characteristics they believe...”. I think a comma is missing again between ‘respectively’ and ‘which’.  

Round 2

Reviewer 1 Report

The changes made have definitely improved the paper. Thank you for your work.

Author Response

We are glad that you noticed an improvement in the paper after the revision. Thank you again for your time and effort in helping us revise the article.